# Targeted Delivery to Tumors: Multidirectional Strategies to Improve Treatment Efficiency

**DOI:** 10.3390/cancers11010068

**Published:** 2019-01-10

**Authors:** Olga M. Kutova, Evgenii L. Guryev, Evgeniya A. Sokolova, Razan Alzeibak, Irina V. Balalaeva

**Affiliations:** 1The Institute of Biology and Biomedicine, Lobachevsky State University of Nizhny Novgorod, 23 Gagarin ave., Nizhny Novgorod 603950, Russia; kutovaom@gmail.com (O.M.K.); eguryev@ibbm.unn.ru (E.L.G.); malehanova@mail.ru (E.A.S.); razanzybak@gmail.com (R.A.); 2The Institute of Molecular Medicine, I.M. Sechenov First Moscow State Medical University, 8-2 Trubetskaya str., Moscow 119991, Russia

**Keywords:** targeted drug delivery, cancer treatment, EPR-effect, passive targeting, cancer-specific molecular targets, active targeting, cell-mediated targeting

## Abstract

Malignant tumors are characterized by structural and molecular peculiarities providing a possibility to directionally deliver antitumor drugs with minimal impact on healthy tissues and reduced side effects. Newly formed blood vessels in malignant lesions exhibit chaotic growth, disordered structure, irregular shape and diameter, protrusions, and blind ends, resulting in immature vasculature; the newly formed lymphatic vessels also have aberrant structure. Structural features of the tumor vasculature determine relatively easy penetration of large molecules as well as nanometer-sized particles through a blood–tissue barrier and their accumulation in a tumor tissue. Also, malignant cells have altered molecular profile due to significant changes in tumor cell metabolism at every level from the genome to metabolome. Recently, the tumor interaction with cells of immune system becomes the focus of particular attention, that among others findings resulted in extensive study of cells with preferential tropism to tumor. In this review we summarize the information on the diversity of currently existing approaches to targeted drug delivery to tumor, including (i) passive targeting based on the specific features of tumor vasculature, (ii) active targeting which implies a specific binding of the antitumor agent with its molecular target, and (iii) cell-mediated tumor targeting.

## 1. Introduction

A personalized approach has emerged a few decades ago as a rapidly developing paradigm of disease treatment. The shift to personalized medicine is most clearly represented in oncology: according to the Personalized Medicine Coalition (PMC) reports 2014–2017 approximately 50% of lately FDA approved novel personalized medicines are antitumor agents. The progress of the approach is based on accumulation of large amount of data on the molecular basis of carcinogenesis and tumor growth regulation as well as architectural peculiarities of tumors tissue [1]. Structural and molecular features of tumors provide an ability to directionally deliver antitumor drugs with minimal impact on healthy tissues and reduced side effects. The overall strategy of personalized treatment implies histological analysis and molecular profiling of an individual tumor, potential targets characterization and selection of a personal treatment tactics. The additional testing on patient-derived 3D-in vitro or in vivo tumor models can be included to experimentally confirm the potency of the chosen drug or drugs combination [2,3,4]. 

Among the features which characterize neoplastic lesions are disordered (compared to normal) tissue structure and aberrant vasculature mostly due to a discrepancy in the tumor mass growth rate and the rate of formation of new blood and lymphatic vessels. This causes the tumor cells deprivation in nutrient and oxygen and elevation of tumor interstitial pressure. Hence hypoxic and acidic conditions develop in the tumor tissue and the extracellular matrix undergoes a transformation towards compaction/stiffness which promotes further tumor progression [5,6,7]. These alterations in tissue architectonics, microenvironment and vasculature become the selective criteria which distinguish the normal and malignant tissues, and thus they can be used as a basis for rational drug design and targeted delivery to tumor site.

Tumor targeting effectiveness can be enhanced by taking into account an altered molecular profile of malignant cells. Tumors are capable of a massive production of molecular factors which condition tumor progression and invasion (growth factors, interleukins, proteases of different classes, etc.) and are characterized by overexpression of G-protein-coupled receptors and receptors of growth factors, transferrin, some interleukins, vitamins, and sugar moieties. The presence of the characteristic molecules on the tumor cells surface and in tumor microenvironment can serve as a specific label to targeted drugs functionalized with specific targeting moiety, leading to increased drug accumulation in the tumor [8,9,10].

An alternative relatively new approach is based on using cells which possess a tropism to tumors as carriers for a therapeutic agent. The loaded cells can migrate along with the gradients of molecules acting as attractants e.g., cytokines, chemokines or growth factors. The promising candidates for cell-based drug delivery are immunocytes. The main benefits of this approach are drug delivery through blood–tissue barriers, hiding the agent from the cells which are responsible for drug clearance and metabolizing; and prolonged circulation in an organism [11].

It should be noted that the targeted treatment implies also using a wide range of agents that do not exhibit a specific tropism to tumor cells, but have a specific action towards them. In particular, tyrosine kinase inhibitors can be attributed to this group. In the present work we do not consider this type of agents. 

In this review we summarize the information on the diversity of currently existing approaches to targeted drug delivery to tumor, including (i) passive targeting based on the specific features of tumor vasculature, (ii) active targeting which implies a specific binding of the antitumor agent with its molecular target, and (iii) cell-mediated tumor targeting.

## 2. Passive Targeting

### 2.1. Tumor Vasculature Peculiarities

Passive targeting is associated with the structural features of the tumor vasculature (Figure 1). The formation of new vessels that provide tumor nutrition and metabolite outflow is a vital condition for tumor development. This process starts when the tumor reaches about 1–2 mm in size [12,13,14]. The aggressive growth of malignant cell population is accompanied by a massive release of pro-angiogenic factors, which leads to the chaotic growth of newly formed vessels. The tumor vascular bed is comprised of immature and tortuous vessels and is characterized by a disordered structure with no defined vessel hierarchy and no clearly distinguishable arterioles, capillaries, and venules [15]. Tumor vessels are irregular in shape and diameter; they possess uneven inner surface with abnormal protrusions, blind ends, arteriovenous shunts, and so-called plasma channels that lack blood cells [16,17]. 

The newly formed lymphatic vessels also have aberrant structure: they are dilated and their endothelial lining is discontinuous and leaky. The integrity of tumor lymphatic vessels is often more disrupted than that of blood vessels. This might be due to better mechanical resistance of blood vessels to the excessive interstitial pressure of the tumor, conditioned by a residual arterial pressure [18,19]. Lymphatic vessels play an important role in tumor progression and metastasis. Although intratumoral lymphatic network is nearly absent, solid tumors tend to develop an extensive surrounding network of lymphatic vessels. As the tumor mass is poorly drained, an excessive interstitial pressure induces a backflow of exfoliated tumor cells up to the lymph nodes, causing distant metastases, which are associated with poor prognosis [20].

It is also important to mention that advanced tumors often form vessel-like structures lined by tumor cells, in order to overcome severe hypoxia and nutrient deprivation. This phenomenon is known as vasculogenic mimicry and it is associated with poor prognosis and low survival rates [21,22].

The vascular network architectonics and geometric resistance to blood flow determine significantly impaired functionality of the tumor vasculature: the decreased blood flow rate, inadequate oxygen supply and subsequently the development of hypoxia at the micro-regional level [23,24,25,26]. The discontinuous endothelial lining of tumor vessels with the lack of smooth muscle layer and impaired integrity of the basement membrane conditions an increased vascular permeability which leads to the release of vascular contents into the intercellular space and to an elevation of interstitial pressure in the tumor, which in turn provokes a collapse of the vessels and insufficient perfusion of the tumor [27,28].

The gaps between endotheliocytes and fenestrations in tumor vessels can vary in diameter from 200 nm to 2 μm and more [29,30], whereas in most normal tissues they do not exceed 5 nm [31]. 

In addition to the inconsistency between the rates of tumor mass increase and vascular growth there are a number of factors which are involved in vascular permeability regulation in tumors. Among them are the kinin-kallikrein system [32,33], nitric oxide NO [34,35], reactive oxygen and nitrogen species, including H_2_O_2_, O_2_^–^•, ONOO^–^, formed in various reactions both by tumor cells and tumor infiltrating leukocytes [36,37], carbon monoxide CO, which is a product of the reaction of hemoxygenase [38], prostaglandins [39], matrix metalloproteinases [40,41], protein factors, including vascular endothelial growth factor (VEGF), transforming growth factor TGF-β, tumor necrosis factor TNF-α [42,43,44]. The tumor development and tumor-associated inflammation typically shift the balance of the listed systems towards an increase in vascular permeability, which is a factor contributing to the intravasation and dissemination of tumor cells.

### 2.2. EPR-Effect in Drug Delivery to Tumor

Both structural and functional features of the tumor vasculature determine the relatively easy penetration of large molecules, supramolecular complexes, as well as any nanometer-sized particles independently of their nature, through the blood–tissue barrier and their accumulation in the tumor tissue. This phenomenon was first described by Y. Matsumura and H.A. Maeda and called the Enhanced Permeability and Retention effect (EPR effect) [45]. The idea of using dimensional effects for selective delivery caused a wide response in the research community (see reviews: [46,47,48,49,50,51]).

To date, the EPR effect has been shown for a wide range of agents e.g., liposomes, micelles, nanodisperse albumin and its modifications, polymer nanoparticles, dendrimers, inorganic nanoparticles of different composition. Some of these agents such as liposomes, micelles, and nanodisperse albumin modifications are approved for clinical practice or undergo different phases of clinical trials, other types of agents are being intensively investigated. A brief overview of them is given below.

Liposomes are the nanoscale vesicles composed mainly of phospho- and sphingolipids which are organized into a bilayer and enclosed into a ball-like structure. To date liposomal antitumor drugs approved by FDA are represented by liposomal forms of doxorubicin (Doxil^®^(ALZA corp., Mountain View, CA, USA), Caelyx^®^ (Janssen Inc., Toronto, Canada), Myocet^®^ (Teva B.V., Haarlem, The Netherlands)) used for treatment of ovarian and breast cancer, multiple myeloma and Kaposi’s sarcoma [52,53], daunorubicin (DaunoXome^®^ (Galen Ltd., Craigavon, UK)) for treatment of HIV-associated Kaposi’s sarcoma [54], cytarabine (Depocyt^®^ (Pacira Pharmaceuticals, Inc., San Diego, CA, USA)) for treatment of neoplastic meningitis, mifamurtide (Mepact^®^ (Takeda Pharmaceutical Co. Ltd., Tokyo, Japan)) for treatment of high-grade, resectable, non-metastatic osteosarcoma, vincristine (Marqibo^®^ (Spectrum Pharmaceuticals, Inc., Henderson, NV, USA)) for treatment of acute lymphoblastic leukemia and irinotecan (Onivyde® Merrimack Pharmaceuticals, Inc., Cambridge, MA, USA) for treatment of metastatic pancreatic adenocarcinoma. Clinical trials are carried out on liposomal forms of doxorubicin and its analogues, cisplatin, topoisomerase I (topotecan, lurtotecan and irinotecan metabolite) and II (mitoxantrone) inhibitors, taxanes (paclitaxel and docetaxel), siRNAs, and others. A comprehensive review on liposomal antitumor formulations is provided elsewhere [55,56]. The further directions in antitumor liposomal formulations progress are the development and improvement of stimuli-sensitive smart liposomes (thermo-, redox-, ultrasound-, enzyme-sensitive), magnetic liposomes, and liposomes for photodynamic therapy [57,58,59,60,61,62,63].

A promising nanoscale drug carrier is the nanodisperse albumin. This protein possesses an outstanding binding capacity due to its molecular structure [64]. Thus it is possible to modify the properties of albumin-based nanoparticles using surfactants, cationic and thermosensitive polymers, PEG, or targeting moieties such as folate, transferrin, apolipoproteins, peptides, or mAbs. [65]. The positive example of using albumin nanoparticles in antitumor treatment is confirmed by a successful application and FDA approval of Abraxane^®^ (Celegene Corporation, Summit, NJ, USA) (paclitaxel incorporated in albumin nanoparticles for a metastatic breast cancer, non-small cell lung carcinoma, pancreatic cancer, cervical cancer [66,67,68,69,70]. Albumin based nanoparticles loaded with lapatinib (triple negative breast cancer and HER2-positive breast cancer), gemcitabine (pancreatic cancer), siRNAs (breast and lung cancers), gold nanoparticles for NSCLC, photosensitizers are also being developed [71,72,73,74,75,76,77].

Organic polymer particles are a promising alternative to liposomes or albumin. An advantage of using such type of particles is in the rational molecule design that allows highly reproducible synthesis of the carriers with desired properties. There is a variety of intensively studied drug carrying vehicles which can be designed with synthetic (polyethylene glycol, polylactic and polyglycolic acids, polycaprolactone, derivatives of polymethacrilic acid, etc.) [78,79,80,81,82,83] or natural polymers (gelatin, collagen, chitosan, dextran, solid lipid nanoparticles, etc.) [84,85,86,87]. Besides the material-based diversity of the polymer nanoparticles, there is also a diversity in shape, size, surface charge and structural organization [88,89,90,91,92,93].

To combine targeted drug delivery with a possibility to use the agent for diagnostic purposes, the inorganic nanoparticles of different composition can be applied. Thus, noble metals are now widely used to create tumor targeting agents. Golden, silver and platinum nanoparticle delivery systems showed pronounced antitumor efficacy towards in vitro and in vivo tumor models [94,95,96,97,98] and represented themselves as a potent tumor imaging agents [99,100]. Highly fluorescent quantum dots, up-conversion nanoparticles, carbon-based nanoparticles are the promising tool for fluorescence guided drug delivery [101,102,103,104,105]. To engage the magnetic resonance imaging, superparamagnetic nanoparticles of iron oxides can be employed [106,107,108,109]. Porous inorganic materials are also being developed as containers for therapeutic or imaging compounds [102,110,111,112]. The above list of organic and inorganic nanoparticles is far from complete and the range of nanoparticles used for the drug delivery is constantly expanding.

As mentioned above passively targeted agents accumulate in the tumor tissue due to its peculiar structure and architectonics. In this respect a fact must be mentioned that nanoscale agents can also be accumulated by tumor associated macrophages which act in this case as a reservoir, gradually releasing the active substance, which significantly prolongs the presence of the agent in the tumor [113].

### 2.3. Questioning the EPR Efficiency

Despite the generally accepted approach of using the EPR effect for targeted delivery to a tumor, it is worth noting, however, that this effect is well manifested in rapidly growing experimental tumors in animals and cannot be considered common for all solid tumors by default. For example, a meta-analysis of preclinical data collected over 10 years showed that the median value of the drug dose delivered to a tumor when using nanoscale carriers is only 0.7% of that was administered [114]. This value several times exceeds the delivery efficiency of low-molecular analogs, which, as a rule, is no more than 0.2%, but nevertheless cannot be considered as fully satisfying the requirements of clinical oncologists.

Though the multiple reports about successful EPR effect-based creative solutions in cancer treatment are expanding it should be noted, that data are accumulated on the absence of the expected effect when moving from animal models to human tumors [115,116]. The most evident example illustrating the gap between preclinical models and clinical practice is the clinical trials results for FDA-approved nano-formulations Doxil^®^ (PEGilated liposomal doxorubicin) and Abraxane^®^ (nanoparticle albumin-bound (nab)-paclitaxel). The treatment efficiency of Doxil^®^ was comparable to free doxorubicin. Its clinical benefit resulted from the modified profile of side effects, primarily from reduced cardiotoxicity [117]. The second example, Abraxane^®^, in clinical trials was significantly more efficacious than conventional formulation Taxol. In this case, the emulsion of Cremophor^®^ EL (currently known as Kolliphor^®^ EL (BASF SE, Ludwigshafen, Germany)), a modified castor oil used for paclitaxel solubilization in Taxol, was eliminated, that resulted in significant increase of maximum tolerated dose [118]. In both cases, the benefits over conventional formulations can be credited to EPR-effect only to a small extent. 

Currently, the inconsistency in drug distribution in animal tumor models and in tumors of human patients is considered to be a potential cause of such a discrepancy in the results of preclinical and clinical trials. The growth rate of a human tumor and its relative volume is much smaller than in case of an experimental model, at the same time an absolute volume is often much larger. Thus, the microenvironmental conditions in the human tumors have their peculiar characteristics: vascular network with fewer fenestrations, hypoxic and acidified sites formed due to the heterogeneity of the blood supply, heterogeneous basal membrane and reduced pericyte coverage, matrix heterogeneity and rigidity, elevated interstitial fluid pressure which promotes convective transport with subsequent nanomedicine clearance from the central zone of the tumor to its periphery [119,120]. The set of these parameters can vary considerably basing on individual characteristics of the patient, type of the tumor the patient suffers of and the stage of its development, hence the manifestation of EPR effect also differs [121].

Optimization of the carrier size is the simplest and promising approach of improving drugs targeting to tumors, but it demands further development. The following directions seems to be the most applicable: elaboration of the tools for controlled modification of vascular permeability (see review: [50]); systematic study on the dependence between physicochemical properties of the carrier and drug delivery efficiency, which would allow the rational design of agents with desired properties; creation of the personalized treatment approaches with consideration of the histomorphological features of the individual tumor (as an example: [121]); and elaboration of approaches combining the use of the EPR effect with other methods of targeted delivery.

## 3. Active Targeting

Active targeting is an approach complementary to passive targeting and is aimed at improvement of accumulation selectivity and increased time of intratumoral retention of the antitumor agent. The active targeting strategy implies covalent or non-covalent binding of the antitumor agent to the molecule, which is capable of selective interaction with specific molecules on the surface of target cells (Figure 2). Active targeting of oncological lesions is possible due to altered molecular profile of malignant cells which is resulted from significant changes in tumor cell metabolism (compared to normal cells) at every level from the genome to metabolome [122,123,124]. We should emphasize that this approach manifests its targeting properties only at microscale: the targeted agent should be located no further than 0.5 nm from the target [125]. The previous drug route from the administration into the body to tumor site is directed by other mechanisms.

### 3.1. Cancer-Specific Molecular Targets

In order to precisely attack a tumor an appropriate target should be selected. A target molecule must be overexpressed on the surface of malignant cells compared to normal cells or, in ideal situation, be absent in normal tissues (Table 1). Also, the potential using of tumor stroma associated targets should be noted.

EGFR (HER1-4) epidermal growth factor receptor family is the most studied in terms of involvement in tumor metabolism and targeting fitness. Overexpression of receptors of this family is characteristic for many types of carcinomas, including breast, lung, stomach, brain and some other cancers [165,166]. EGFR/HER1, HER2 and HER3 receptors are successfully used as targets for targeted delivery of various antitumor agents [167]. To date there is only one FDA approved anti-tumor agent targeted to the EGFR family member, excluding non-conjugated therapeutic antibodies from consideration. Kadcyla^®^ is based on an antibody specific for HER2 (Trastuzumab) and an antimitotic agent emtansine (DM1) inhibiting the polymerization of microtubules that are chemically linked together via a non-reducible thioether linker. This agent is used for the second line treatment of HER2-positive metastatic breast cancer.

Clinical and preclinical trials of Kadcyla^®^ are underway for other types of HER2-overexpressing tumors, for example, for monotherapy of metastatic breast cancer and for therapy of gastric cancer in patients previously treated with taxane drugs [168].

Chemical conjugation allows obtaining targeted agents of various specificity with different toxic/cytostatic moieties. For example, this approach is used to create EGFR-specific agents for boron neutron capture therapy of brain tumors, which are now undergoing preclinical trials [169,170]. Chemical conjugation of functional modules has its drawbacks such as manufacturing difficulties and varying composition. Progress in genetic engineering has allowed the creation of recombinant antitumor agents where protein toxins of various origins and the mechanism of action are fused with a targeting moiety into a single polypeptide chain. The wide spectrum of recombinant antitumor agents specific to EGFR family receptors undergo clinical [171,172,173,174] and preclinical trials [131,135,136,175,176,177,178,179,180,181,182,183,184,185].

Tumor development is accompanied by angiogenesis which is naturally associated with the high expression of vascular endothelial growth factor receptors (VEGFR 1–3) and integrins (αvβ3 and others), mainly on endothelial cells [186]. The applicability of these receptors as target molecules is twofold: they can be used for targeted drug delivery and as the direct means for the tumor disruption by restriction of nutrients supply [187,188,189]. VEGF-receptors are often present in a prostate cancer, melanoma and leukemia. The examples of other growth factor receptors associated with tumor development and used as targets are platelet growth factor receptor PDGFRα/β [142] and insulin-like growth factor receptor IGF-1R [144].

Rapidly dividing cancer cells are in a great need for iron so the increased expression of transferrin (iron-binding protein) receptors, TfR1-2 is reported for brain, lung, bladder, intestine, pancreas, and some other cancers [190,191]. Thus targeted drug delivery can be performed using both transferrin itself and antibodies to its receptor as a targeting moiety [191,192,193]. Similarly, a significant proportion of fast-growing tumors are characterized by high expression of receptors for folic acid [194,195], biotin [196,197], and other vitamins, as well as membrane carriers of sugars [198,199,200].

In case of blood cancers, including various forms of lymphoma, leukemia and myeloma, the lymphocyte antigens and a number of other proteins which are overexpressed on the surface of transformed cells are employed as molecular targets. Disseminated nature of such tumors promoted the implementation of the targeted approach to treatment. To date, a wide number of molecular targets, including clusters of differentiation CD3, CD19, CD20, CD22, CD25, CD27, CD30, CD33, CD37, CD40, CD52, CD56, CD70, CD74, CD79, CD80, CD138, CD 307, and B-cell maturation antigen BCMA, and some other have been tested (with varying success) for the drug delivery to hematological tumors (for details see reviews: [201,202,203]).

The range of potential targets has significantly expanded in recent years. Thus, it has been shown that aberrant expression and activation of a number of members of G-protein-coupled receptor family plays an important role in carcinogenesis, tumor growth and invasion, cell migration and metastasis.

The most studied proteins of this group are receptors of angiotensin, lysophosphatidic acid, sphingosin-1-phosphate, melanocortin, vasopressin, estrogen, gastrin-releasing peptide, etc. [9,204]. This extensive group of receptors is of great interest and, apparently, the number of targeted agents specific to them will rapidly increase in the nearest future.

Several other molecular targets ought to be mentioned: interleukin receptors expressed, particularly, in some types of gliomas [205]; mesothelin which is involved in cell adhesion and is highly expressed on mesothelioma cells and in a number of adenocarcinomas [206,207]; prostate-specific membrane antigen PSMA [208,209]; plasma membrane proteoglycans, for example the mucin (MUC-1) overexpressed in carcinomas [210,211]. The altered glycosylation profile of the tumor cells surface makes them possible to be recognized by lectins [212].

The above listed molecules are expressed on the surface of tumor cells. However, stromal components of a tumor, both nonmalignant cells and extracellular matrix (ECM), can also serve as a target for directional treatment. Tumor progression is strongly determined by the microenvironment of tumor cells and crosstalk between tumor cells, cancer associated fibroblasts, endothelial cells, pericytes, and smooth muscle cells composing tumor vasculature, as well as infiltrating immune and inflammatory cells [213]. Targeting non-malignant cells of tumor stroma (including their signaling pathway participants) [214,215,216] and vasculature [217,218,219] seems to be the potent treatment approach with most remarkable progress in immune checkpoints inhibition [220].

To date, the existing treatment strategies are mostly aimed at improvement in blood perfusion, drug extravasation and tissue penetration by modulating of tumor stroma as well as at activation of the antitumor immune response [221,222]. However, the specificity of tumor microenvironment makes it potentially possible to develop the approaches for drug delivery based on targeting stromal compartment. For example, a number of antiangiogenic nanoparticle-based agents with receptor specific peptides have been reported [223,224].

The list of aforementioned cancer-specific molecular targets is far from complete. The development of molecular oncology and the gradual deciphering of the mechanisms responsible for cells transformation and regulation of their malignant phenotype lead to the formulation of new principles and approaches to control tumor progression and eliminate tumor cells. It can be said that currently there is an active search for new targets that would provide effective antitumor treatment [225,226].

### 3.2. Targeting Agents

Antibodies and their recombinant derivatives are target-recognizing molecules most commonly used for drug delivery to tumor [227,228]. Historically, the first antibody fragment were obtained by hydrolytic cleavage of a full-size antibody and included one or two antigen-binding domains (Fab and (Fab)2, respectively) [229]. The development of recombinant protein technology lead to creation of scFv-type (single chain fragment variable) fragments represented by variable domains of the heavy and light chains fused together by a peptide linker [230,231], and dsFv-type (Disulfide-stabilized fragment variable) fragments in which the variable domains are linked by a disulfide bond [232,233]. Single-chain antibodies of the Camelidae family and Chondrichthyes class were used to construct single-domain antibodies (sdAb or nanobodies) [234]. The major advantage of recombinant antibody fragments is the possibility of their genetic fusion with other proteins in order to create multivalent and/or multispecific agents [235,236,237]. A plethora of targeted antitumor agents based on antibodies derivatives are under preclinical efficiency trials. There are some promising results obtained for scFv-based [128,183,238], dsFv-based [148,239,240], nanobody-based [184,241] immunotoxins. Antibody fragments are also widely used as a targeting moiety of hybrid multifunctional complexes with QDs, superparamagnetic iron-oxide, gold nanoparticles, etc. [101,107,242].

Although antibody-based agents are indisputably potent drug targeting units there is a promising alternative represented by non-immunoglobulin scaffold proteins which also have hypervariable sites responsible for highly specific target recognition [243,244,245]. These proteins possess several advantages compared to antibodies: they have no propensity to aggregate and commonly are highly thermodynamically stable so they are less susceptible to the external factors such as temperature, pH, and protease activity [246]. The valuable characteristic of non-immunoglobulin scaffold proteins is their smaller size. Targeting moieties based on adnectins, affibodies, DARPins, and knottins are successfully tested [132,139,141,143,247,248,249]. The next promising group is represented by fynomers [250,251], anticalins, and Kunitz domains (modified Kunitz-type inhibitors) which demonstrated high binding capacity and can be potentially applied for targeted delivery of various agents [252,253].

In the case when the molecular target is represented by membrane receptor, the delivered agent can be conjugated directly to the ligand of this receptor. This approach is widely used to antitumor agent design both in case when the targeting moiety is a low molecular weight compound (folate, carbohydrate residues, vitamins etc.) [147,194,254] or a protein (transferrin, growth factors) [131,255].

Finally, it is worth mentioning peptides as a means of targeted drug delivery. These peptides may be divided into two groups. The first group is represented by tumor-homing peptides (THPs) which are capable of specific recognition of certain tumor-associated antigens. The best-known THP is RGD (Arg-Gly-Asp) binding to integrins ανβ3 and ανβ5, which are widely present on the surface of endothelial cells of tumor vessels [256]. This sequence was first discovered as minimal binding epitope of fibronectin [257]. To date two forms of RGD peptides are being investigated: linear tri-heptapeptides and cyclic motifs. The structural rigidity of cyclic RGD peptides conditions their elevated specificity to receptor subtypes compared to linear peptides, they are also less susceptible to degradation [258,259]. Internalizing RGD (iRGD) peptides with 9-amino acid cyclic motifs was shown to launch transcytosis of nano-carriers through the vascular endothelium. This allows delivering loaded nano-carriers to the tumors with non-leaky vasculature where the EPR based delivery strategy is not applicable. A significant improvement of drug delivery and hence antitumor efficacy has been shown for a wide variety of nano-carriers in multiple tumor models [260], both in case when the iRGD-peptide was chemically bound to the nano-carrier [261,262] or co-administered [263].

Among cell penetrating peptides chlorotoxins or CTX-like peptides (a group of peptides derived from the scorpion venom) is intensely studied as targeting agents for targeted drug delivery. CTX peptide was purified and characterized in early 1990s [264] and then was shown to specifically bind to glioma cells [265]. The further research in this area indicated that CTX-like peptides can be internalized both by the energy-dependent (endocytosis) and by the energy-independent way. It was also shown that CTX-like peptides have a tumor binding activity, but their molecular target is not yet specified; nevertheless chlorine channels, matrix metalloproteinase-2 and annexin A2 are considered as potential candidates. Currently, the use of CTX-like peptides is investigated for various types of tumors (mainly for brain tumors) for imaging, drug delivery and radiotherapy [266,267].

Another representatives of the THP group are transferrin receptor-specific peptides (7pep, HAIYPRH), peptides having R/KXXR/K motif, which undergo neurophilin mediated internalization, and many others, including artificially synthesized peptides [268,269].

The second group of peptides with high potential for targeted drug delivery is represented by cell-penetrating peptides (CPPs) which are non-homologous peptides composed with 5–30 a.a. and originated from different organisms including humans. The common property of these peptides is the ability to penetrate the cell membrane by endocytosis or direct translocation. The penetration mechanism is not yet fully comprehended, but obviously is associated with the presence of cationic amino acid residues in the peptides. Cell-penetrating peptides have no binding selectivity to cells of a specific molecular profile but significantly facilitate cell entry [270,271].

Thus, active drug delivery to tumor cells is an intensively developing approach. Active search for new targets, development of targeting moieties specific to them and employing of various toxic and/or imaging moieties led to an explosive growth of a number of candidate antitumor agents in recent years. The spectrum of delivered agents extends from cytostatic drugs such as DM-1, doxorubicin, maytansinoid [140,147,155], truncated plant and bacterial toxins e.g., gelonin and *Pseudomonas* exotoxin A [128,131], radionuclides mostly for imaging purposes (^99^Tc, ^111^In, ^68^Ga) [133,138], up to photosensitizing dyes e.g., porphyrin, phthalocyanine [272,273] and nanoparticles of different nature (QDs, carbon, polymer, gold nanoparticles, UCNPs) [101,139,144,151,152,272]. The latter group, targeted nanoparticle-based drugs, is particularly attractive for producing hybrid multifunctional complexes combining properties of imaging and therapeutic agent.

## 4. Cell-Mediated Targeting

In recent years, a new approach has been proposed implying drug delivery by cells which possess preferential tropism to tumor (Figure 3).

This approach possesses some distinct advantages: it allows active delivery of the loaded drug directionally to the target site, prolonged half-life, gradual and controlled release and decreased side cytotoxicity and immunogenicity [11]. Certain cell populations are able to infiltrate a tumor despite an increased interstitial pressure and the presence of a tumor stroma. Gradients of cytokines (macrophage colony-stimulating factor CSF1, pro-inflammatory cytokines), chemokines (particularly those which are recognized by the CXCR4/CXCL12 receptor system, as well as the MCP-1 monocyte chemotactic protein) and growth factors (VEGF, TGF-β and fibroblast growth factor FGF-2) can act as cell attractants [274]. To date, several cell types have been tested as drug carriers. Thus, naive T-cells tropic to lymph nodes were successfully used to attack tumors of this localization [275]. Primed T-cells specific to a certain tumor cell surface antigen can be used in case of tumors of other localizations [276]. In addition, cytokine and growth factor gradients direct the migration of some cell types, which allows using them as drug carriers. Monocytes and neutrophils [277,278,279], macrophages [280,281], as well as mesenchymal stem cells from bone marrow and cord blood [282,283], neural stem cells [284,285] and some other cell types were successfully applied for antitumor agents delivery.

The procedure is basically associated with the collection of autologous or donor material, ex vivo cell loading/activation, expansion the cells to the required quantities and administration to the patient. Cell-mediated approach provides delivery of low-molecular compounds, proteins, genetic material, nanoparticles and oncolytic viruses possible [286,287,288]. Recently developed T-cell genetic “reprogramming” using CAR (chimeric antigen receptor) technology producing cells with designed specificity to antigens and intended to activate antitumor immunity (see review: [289]) seems to be also promising when combined with cell loading by antitumor agents.

Although a significant success was achieved in this area, there are still some limitations which ought to be considered and overcome. They are associated both with the method procedure and specific properties of the cells acting as drug carriers. Methodological limitations include the risk of the carrier-cells contamination during cultivation and loading with possible subsequent blood contamination, drug loading difficulties e.g., low loading capacity and disintegration within the carrier-cell, limited control of drug release. Cell type specific limitations are generally represented by short ex vivo life, impaired resistance to damaging factors (mechanical or osmotic) induced by loading procedures and cultivation difficulties. Thus, platelets are prone to induce thrombogenesis, leucocytes are characterized by poor transduction level, stem cells tend to lose potency in vitro [290]. It must be noted that many technologies from this group are directly related to the production of tumoricidal cells.

## 5. Conclusions

The rapid accumulation of knowledge on the mechanisms driving carcinogenesis and peculiar features of tumor growth lead to development of a number of approaches to the targeted delivery of therapeutic agents to tumor cells or to the newly formed tumor vasculature. The opportunity of choosing the most appropriate treatment method which takes molecular and histomorphological tumor features into account and precise adjustment of administration schedule and regime is moving us towards ideals of personalized medicine.

Despite the significant progress in the area and the creation of an unprecedentedly wide range of targeted agents which are now at the clinical or pre-clinical trials, it is necessary to admit that there is still a huge range of outstanding issues. Thus methods to increase passive targeting efficiency and reduce drug dissipation (an unwanted capture by liver or kidney excretion) have to be developed. There is a direct relationship between half-time of blood circulation and the efficiency of passive delivery. The commonly proposed coating of the delivered agent with poly(ethylene)glycol (PEG) or its analogs has now been being critically revised. Such type of coating is intended to prolong circulation time by prevention of protein opsonization on the agent surface and following capturing of the agent by macrophages [291]. However, about 30 years of clinical experience resulted in elucidating possible side effects and complications, including registered cases of hypersensitivity, unexpected effects on drug pharmacokinetics, non-biodegradability and possible accumulation in the body, toxic side products, and production of anti-PEG antibodies [292,293,294]. Nowadays the problem of a protein corona formation on the nano-sized agents and its impact on a biological identity including specificity to target cells and retention in pathological sites is among the highly attractive for research community. The proposed ways to optimize blood circulation time and enhance the delivery efficiency vary from using low adhesive coatings which prevent protein binding [295,296] to preconditioning of the delivered agent with proteins such as albumin before they are administered in to the blood stream to artificially produce protein corona with desired prosperities [297,298].

Another perspective approach is to improve the delivery efficiency is to affect the tumor microenvironmental processes, especially angiogenesis and oxygenation. Antiangiogenic drugs block invasion of new vessels into the tumor and can temporarily normalize tumor vasculature and hence the oxygenation level. This leads to a reduction of tumor interstitial pressure, improved drug delivery, and the realization of the “oxygen effect” during radiotherapy [299,300].

The common limitation of drug delivery to solid tumors is a poor penetration of the any agents into the deep of the tumor mass [301]. The approach has been proposed to improve the penetration efficiency by simultaneous targeting the tight junctions which are widely represented in solid tumors. Several protein agents targeting intercellular tight junctions has been created and demonstrated its potency in combination with various agents [302,303,304]. This approach is of particular interest when applying targeted agents specifically binding to cell surface receptors, since the latter are often hidden under cell adhesion proteins and may be unavailable for binding.

To summarize, the progress in understanding intratumoral transport of nutrients and metabolites, tumor peculiarities at molecular and cellular level, growing list of potential molecular targets of tumor cells, extending amount of data elucidating tumor interaction with cells of immune system allow to expect generation of novel ideas and solutions to overcome encountered issues and to improve antitumor drug delivery and treatment efficiency.

## Figures and Tables

**Figure 1 cancers-11-00068-f001:**
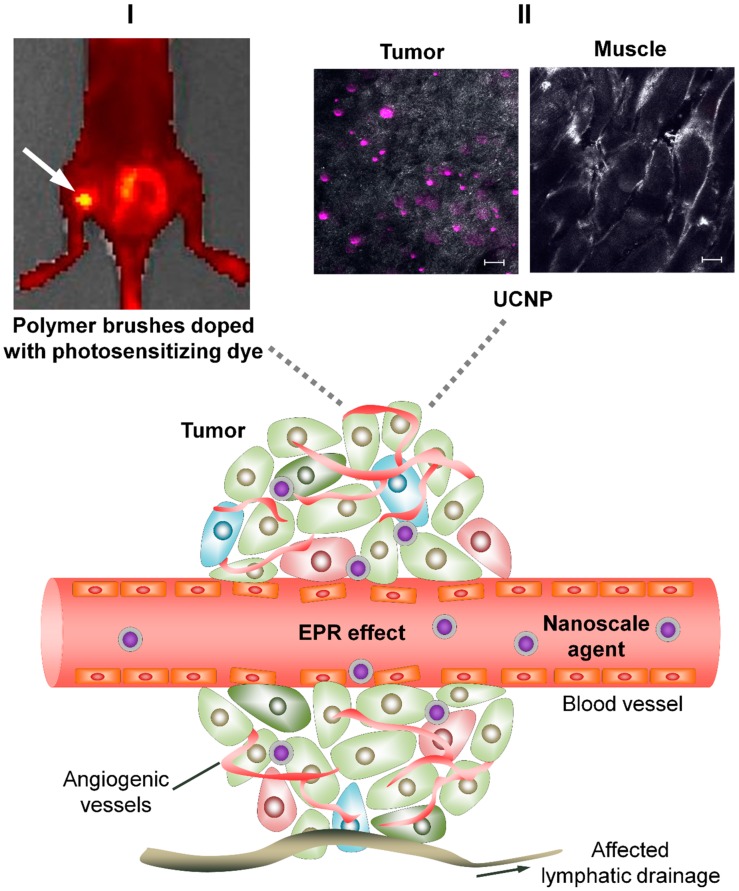
Scheme illustrating the principle of passive drug delivery to the tumor. EPR effect: Permeability and Retention effect; UCNP: upconversion nanoparticles. The extravasation and penetration of nanoscale agents into the tumor is due to the disordered structure of the tumor vessels, their discontinuous endothelial lining and the disrupted integrity of the basement membrane. Irregular diameter and leakage of the walls of the newly formed lymphatic vessels impede the outflow of fluid and the removal of nano-sized agents from the tumor. Insets I and II indicate examples of passive drug delivery in vivo. I—Polymer particles based on water-soluble polymer brushes (polyimide-graft-polymethacrylic acid) are used to deliver photodynamic dye (tetra(4-fluorophenyl)tetracyano porphyrazine). The image was obtained by whole-body imaging 24 h after intravenous injection of a dye to BALB/c mouse with CT26 allograft (murine colorectal carcinoma) in the left thigh. The position of the tumor is indicated by an arrow. The fluorescence intensity of the dye is presented in the gradient red to yellow scale, where yellow corresponds to the maximum signal. II—Passive delivery of upconversion nanoparticles (UCNP) of composition, NaY:Yb:Tm:F4/NaYF4 covered with alternating copolymer of maleic anhydride and 1-octadecene (PMAO). Tumor and muscle tissue images were obtained ex vivo by confocal fluorescence microscopy 3 h after intravenous injection of UCNP-PMAO to a BALB/c mouse with SK-BR-3 xenograft (human breast adenocarcinoma). Purple signal corresponds to the UCNP photoluminescence. Scale bar 20 μm.

**Figure 2 cancers-11-00068-f002:**
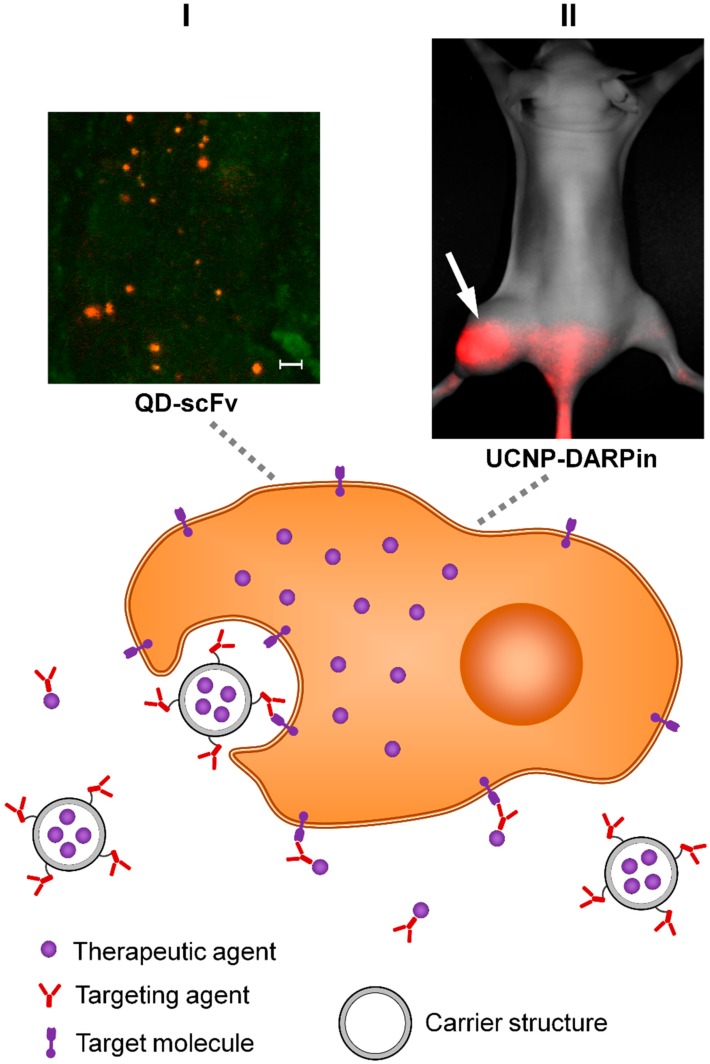
Scheme illustrating the principle of active drug delivery to the tumor. QD: quantum dots; scFv: single chain fragment variable. Active delivery implies covalent or non-covalent binding of the delivered agent to the moiety, which determines its selective interaction with specific molecules on the surface of target cells. This moiety can be attached directly to the delivered drug or to a nano-sized container loaded with a therapeutic drug. Insets I and II indicate examples of active drug delivery in vivo. I—Active delivery of NIR fluorescent quantum dots (QD) bound with anti-HER2 scFv (4D5scFv). Image of tumor tissue was obtained by confocal fluorescence microscopy 21 h after intravenous injection of QD-4D5scFv to BALB/c nude mice with SK-BR-3 xenograft (human breast adenocarcinoma). The red signal corresponds to QD photoluminescence. Scale bar 10 μm. II—Active delivery of upconversion nanoparticles (UCNP) of NaY:Yb:Tm:F4/NaYF4 composition bound with HER2-specific protein DARPin. The image was obtained by whole-body imaging 2 h after the intravenous (tail vein) injection of the nanocomplex BALB/c mouse with SK-BR-3 xenograft (human breast adenocarcinoma). The position of the tumor is indicated by an arrow. The red signal corresponds to the photoluminescence of the UCNP.

**Figure 3 cancers-11-00068-f003:**
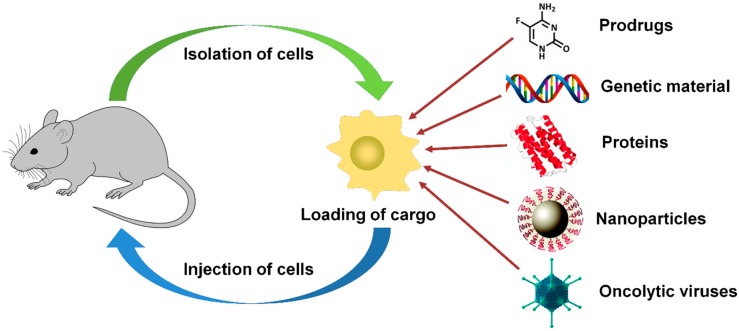
Scheme illustrating the principle of cell-mediated tumor targeting. Drug carriers may be tumor tropic cells: naive T-lymphocytes, primed T-lymphocytes, monocytes, neutrophilic granulocytes, macrophages, mesenchymal stem cells from bone marrow and umbilical cord blood, neural stem cells, and some other cell types. This approach involves the collection of autologous or donor material, loading/activation of the cells under ex vivo conditions, expansion to necessary quantities and introducing them back into the body. Cells can be successfully used to deliver low-molecular compounds, proteins, genetic material, nanoparticles and oncolytic viruses.

**Table 1 cancers-11-00068-t001:** Preclinical and clinical examples of tumor-targeted imaging and drug delivery.

Target	Designation	Targeting Agent	Agent	Patients Group/Animal Model	Reference
**EGFR**	Epidermal growth factor receptor	Cetuximab (Erbitux^®^, Bristol-Myers Squibb Company, New York, NY, USA)	Cetuximab-labeled liposomes loaded with chemotherapy drug Oxaliplatin	Human colorectal cancer xenograft	[126]
		Cetuximab (Erbitux^®^)	Cetuximab conjugated with chemotherapy drug Docetaxel	Human epidermoid carcinoma (A431) xenograft	[127]
		scFv 425	425scFv fused with *Pseudomonas* exotoxin A fragment (recombinant immunotoxin 425scFv-ETA)	Human epidermoid carcinoma (A431) xenograft	[128]
		Nanobody 8B6	^99m^Tc-labeled 8B6 nanobody for SPECT tumor visualization	Human epidermoid carcinoma (A431) xenograft	[129]
		Nanobody D10	^99m^Tc-labeled D10 nanobody for SPECT tumor visualization	Human mammary (MDA-MB-468) and epidermoid (A431) carcinoma xenografts	[130]
		EGF	EGF fused with toxin gelonin (recombinant targeted toxin EGF/rGel)	Human epidermoid carcinoma (A431) xenograft	[131]
**HER2**	Human epidermal growth factor receptor-2	DARPin (HE)3-G3	^111^In-labeled (HE)3-G3 DARPin for SPECT/CT tumor visualization (^111^In-(He)3-G3)	Human breast carcinoma (BT-474) xenograft	[132]
		Nanobody 2Rs15d	^99m^Tc-labeled 2Rs15d for SPECT tumor visualization	Human ovarian carcinoma (SKOV-3) xenograft	[133]
		scFv 4D5	Qdot 705 ITK-labeled 4D5scFv for optical tumor visualization (QD-4D5scFv)	Human breast carcinoma (SKBR-3) xenograft	[101]
		scFv 4D5	4D5scFv fused with *Pseudomonas* exotoxin A fragment (recombinant immunotoxin 4D5scFv-PE40)	Human ovarian carcinoma (SKOV-kat) xenograft	[134,135]
		scFv 4D5	4D5scFv fused with toxin gelonin (recombinant immunotoxin rGel/4D5)	Human ovarian carcinoma (SKOV3) xenograft	[136]
		DARPin9.29	DARPin9.29 fused with *Pseudomonas* exotoxin A fragment (targeted toxin DARPin-PE40)	Human breast carcinoma (SKBR-3) xenograft	[137]
		Affibody ABY-025	^111^In-labeled ABY-025, ^68^Ga-labeled ABY-025	Phase I/II study in patients with breast cancer metastases	[138]
		DARPin9.29	^90^Y-dopped upconversion nanoparticles (UCNP) coupled to targeted toxin DARPin-PE40 (UCNP-R-T)	Human breast carcinoma (SKBR-3) xenograft	[139]
		Trastuzumab (Herceptin^®^, Genetech, Inc., San Francisco, CA, USA)	Trastuzumab conjugated with cytotoxic agent emtansine (DM1) (Trastuzumab emtansine, or T-DM1)	FDA approved for the treatment of patients with HER2-positive, metastatic breast cancer (Kadcyla^®^, Genetech, Inc., San Francisco, CA, USA)	
				Phase II study in patients with previously treated HER2-overexpressing metastatic non-small cell lung cancer	[140]
**HER3**	Human epidermal growth factor receptor-3	Affibody HEHEHE-z08698-NOTA	^68^Ga-labeled affibody HEHEHE-Z08698-NOTA for PET imaging	Human breast (BT-474) and pancreas (BxPC) carcinoma xenografts	[141]
**PDGFR β**	Platelet-derived growth factor receptor beta	Targeting peptides (PDGF, yITLPPPRPFFK)	PDGF-labeled micelles loaded with drug temozolomide (TMZ)	Human glioblastoma (U87) xenograft	[142]
**IGF-1R**	Insulin-like growth factor 1 receptor	Affibody ZIGFR:4551-GGGC	^99m^Tc-Z_IGFR:4551_-GGGC	Human prostate (Du-145) and breast (MCF-7) carcinoma xenografts	[143]
		mAb IGF-IR	Oxidized single-wall carbon nanohorns with incorporated drug vincristine and wrapped with mAb IGF-IR (VCR@oxSWNHs-PEG-mAb)	Mouse hepatoma (H22) syngraft	[144]
**TfR**	Transferrin receptor	Transferrin	Transferrin-labeled liposome–DNA for systemic p53 gene therapy (LipT– pSVb)	Human squamous cell carcinoma of the head and neck (JSQ-3) xenograft	[145]
**PSMA**	Prostate-specific membrane antigen	Nanobody JVZ-007	^111^In-labeled JVZ007 nanobody for SPECT/CT imaging (^111^In-JVZ007-c-myc-his, ^111^In-JVZ007-cys)	Human prostate carcinoma (PC-310) xenograft	[146]
**Carbohydrate moieties**	Oligosaccharides associated with cell membrane lipids, proteins or peptide glycans	Lectin (Bauhinia purprea agglutinin, BPA)	BPA-labeled PEGylated liposomes encapsulating drug Doxorubicin (BPA-PEG-LPDOX)	Human prostate carcinoma (Du-145) xenografts	[147]
**Mesothelin**	Mesothelin	dsFv SS1	SS1dsFv fused with *Pseudomonas* exotoxin A fragment (recombinant immunotoxin SS1(dsFv)PE38)	Phase I study in patients with pleural mesothelioma	[148]
**IL-13Rα2**	Interleukin 13 receptor α2	Linear peptide (CGEMGWVRC, or Pep-1)	Pep-1-labeled PEGylated nanoparticles loaded with drug Paclitaxel (Pep-NP-PTX)	Intracranial rat glioma (C6) xenograft	[149]
**FRα**	Folate receptor α	mAb M9346A	M9346A mAb conjugated with cytotoxic agent maytansinoid DM4 (Mirvetuximab soravtansine, or IMGN853)	Phase I study in patients with advanced, FRα-positive solid tumors (epithelial serous or endometrioid ovarian cancer, primary peritoneal cancer, fallopian tube cancer, serous or endometrioid endometrial cancer, non-small-cell lung carcinoma, and renal cell cancer)	[150]
		Folate	Folate-labeled HEA-b-EHA polymer micelles loaded with drug Orlistat (Fol-HEA-EHA-orlistat NPs)	Human triple negative breast cancer (MDA-MB-231) xenograft	[151]
**Sugar Carriers**	Membrane carriers of sugars	Glucose moieties	D-glucose-labeled fullerene for PDT (C(60)-(Glc)1)	Human melanoma xenograft	[152]
**FSHR**	Follicle-stimulating hormone receptor	Polypeptide of follicle-stimulating hormone (FSHP)	shRNA-loaded FSHP-labeled nanoparticles for blocking growth-regulated oncogene α (gro-α)	Human ovarian carcinoma (HEY) xenograft	[153]
**CD3**	Cluster of differentiation 3	Anti-CD3 scFvs	Two scFvs fused with diphtheria toxin fragment (A-dmDT390-bisFv, or UCHT1)	Patients with cutaneous T cell lymphoma	[154]
**CD19**	Cluster of differentiation 19, or B-Lymphocyte Surface Antigen B4	mAb huB4	huB4 mAb conjugated with cytotoxic agent maytansinoid DM4 (Coltuximab ravtansine, or SAR3419)	Phase II study in patients with relapsed or refractory acute lymphoblastic leukemia	[155]
**CD22**	Cluster of differentiation 22	Inotuzumab	Inotuzumab conjugated with cytotoxic agent calicheamicin (Inotuzumab Ozogamicin)	FDA approved for the treatment of patients with relapsed or refractory B-cell precursor acute lymphoblastic leukemia (Besponsa^®^, Pfizer, Inc., New York, NY, USA)	
				Phase I study in patients with relapsed/refractory CD22+ B-cell non-Hodgkin lymphoma (NHL)	[156]
		anti-CD22 Fv	anti-CD22 Fv fused with *Pseudomonas* exotoxin A fragment (recombinant immunotoxin Moxetumomab pasudotox)	FDA approved for the treatment of patients with relapsed or refractory hairy cell leukemia (Lumoxiti^®^, AstraZeneca PLC, Cambridge, UK)	
				Phase 1 study in patients with acute lymphoblastic leukemia	[157]
**CD25**	Cluster of differentiation 25, or Interleukin-2 receptor alpha chain	anti-CD25 scFv	anti-CD25 scFv fused with *Pseudomonas* exotoxin A fragment (anti-Tac(Fv-PE38), or LMB-2)	Phase II study patients with adult T-cell leukemia	[158]
**CD30**	Cluster of differentiation 30, or TNF receptor superfamily member 8	Brentuximab	Brentuximab conjugated with antimitotic agent monomethyl auristatin E (MMAE) (Brentuximab vedotin)	FDA approved for the treatment of patients with classical Hodgkin lymphoma and anaplastic large-cell lymphoma (Adcetris^®^, Seattle Genetics, Inc., Bothell, WA, USA)	
				Phase I study in patients with mediastinal large B-cell lymphoma	[159]
**CD46**	Cluster of differentiation 46, or Membrane cofactor protein	mAb 23AG2	23AG2 mAb conjugated with cytotoxin agent monomethyl auristatin F (MMAF)	Human multiple myeloma disseminated xenograft (RPMI8226)	[160]
**CD56**	Cluster of differentiation 56, or Neural cell adhesion molecule	Lorvotuzumab	Lorvotuzumab conjugated with maytansinoid cytotoxic agent (DM1) (Lorvotuzumab mertansine, or IMGN901)	Phase I study in patients with CD56-positive relapsed or relapsed/refractory multiple myeloma	[161]
**CD70**	Cluster of differentiation 70, or TNF ligand superfamily member 7	mAb h1F6	h1F6 mAb conjugated with dimeric pyrrolobenzodiazepine (h1F_6239C_-PBD)	Renal cell carcinoma and non-Hodgkin lymphoma xenografts	[162]
**CD74**	Cluster of differentiation 74, or HLA class II histocompatibility antigen gamma chain	mAb hLL1	hLL1 mAb conjugated with drug Doxorubicin (IMMU-110)	Human multiple myeloma (MC-CAR) xenograft	[163]
**BCMA**	B-cell maturation antigen	mAb J6M0	J6M0 mAb conjugated with cytotoxin agent monomethyl auristatin F (MMAF) (J6M0-mcMMAF)	Human multiple myeloma subcutaneous xenografts (H929 and OPM2) and orthotopic disseminated xenografts (MM1S)	[164]

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
