# Peer review of "Targeted Delivery to Tumors: Multidirectional Strategies to Improve Treatment Efficiency"

_cancers, 2019, doi:10.3390/cancers11010068_

Round 1
Reviewer 1 Report
Olga M. Kutova et al. summarized a certain number of studies for targeted delivery to solid tumors. However, targeted nanoparticles for drug delivery were not well included. There are still numerous targeted delivery of drug/gene based on nanoparticles not described in this manuscript. Please add them or give a more specific title rather than this current one. "Targeted delivery to tumours: multidirectional strategies to improve treatment efficiency" would be a title for a book or a journal.
Author Response
Dear Reviewer,
We would like to express our sincere appreciation for your careful attention to our manuscript and for the suggested improvements and valuable comments. We have revised the manuscript according to your remarks. Please find below the detailed description of the revisions (Reviewer’s remarks are presented in bold, our answers follow in plain text, quotations from the manuscript are shown in italic).
Olga M. Kutova et al. summarized a certain number of studies for targeted delivery to solid tumors. However, targeted nanoparticles for drug delivery were not well included. There are still numerous targeted delivery of drug/gene based on nanoparticles not described in this manuscript. Please add them or give a more specific title rather than this current one. "Targeted delivery to tumours: multidirectional strategies to improve treatment efficiency" would be a title for a book or a journal.
The scope of the Review covers the peculiar features of tumour tissue organization and molecular profiling of tumour cells that can be used for targeted delivery. Based on the accentuated peculiarities, we aimed at the discussion of the main approaches to reach the selective drug delivery to tumor.
We absolutely agree with the Reviewer that the title is very wide. The wideness of the title is dictated by the wideness of the scope. The structure of the manuscript implies the notion of the existent approaches including passive, active and cell-based targeting with short but informative descriptions. We do not pretend to systematically review all the huge amount of the targeted agents in clinic and under study at the present moment. Every strategy and concept is illustrated by several relevant examples. Also, every section of the manuscript includes the Refs to not only original papers but to the recent reviews discussing the specific aspects of the targeted delivery and treatment.
In the original version of the manuscript we gave the list of examples of the targeted delivery of nanoparticles in a table form (see Table 1). In the revised version we extended the description of the targeted nanoparticles in the body text.
In the section describing the potency of the antibody-derived targeting modules:
Antibody fragments are also widely used as a targeting moiety of hybrid multifunctional complexes with QDs, superparamagnetic iron-oxide, gold nanoparticles, etc. [101,107,242].
In the section finalizing the description of the active targeting approach:
The spectrum of delivered agents extends from cytostatic drugs such as DM-1, doxorubicin, maytansinoid [140,147,155], truncated plant and bacterial toxins e.g. gelonin and Pseudomonas exotoxin A [128,131], radionuclides mostly for imaging purposes (99Tc, 111In, 68Ga) [133,138], up to photosensitizing dyes e.g. porphyrin, phthalocyanine [272,273] and nanoparticles of different nature (QDs, carbon, polymer, gold nanoparticles, UCNPs) [101,139,144,151,152,272]. The latter group, targeted nanoparticle-based drugs, is particularly attractive for producing hybrid multifunctional complexes combining properties of imaging and therapeutic agent.
Reviewer 2 Report
In this review manuscript by Kutova et al, the authors intended to provide an overview of the strategies that have been developed for targeted drug delivery to tumors. The authors reviewed the biology of tumors relevant to drug delivery, following with discussion of three different approaches. This is a potentially interesting review which may have a broad audience. The following critiques need to be addressed.
1. Emerging evidence shows that the EPR effect, which has been overemphasized based on findings in animal studies, has not been well validated in human patients. The authors need to review the relevant findings in human patients and discuss the potential reasons. This discussion may help distinct this review article from many others with a similar focus.
2. In the Active targeting section, the authors reviewed “Cancer-specific molecular targets” in section 3.1. In addition to the molecular targets expressed on cell surface, molecular targets in tumor microenvironment are also important for drug delivery. The authors need to include a section to provide an overview of these targets.
3. Line 175, “of using synthetic particles is in the rational molecule…” The authors should be aware that liposomes are also synthetic particles.
4. Line 180: “a shape diversity… organized as dendrimers, hyperbranched polymers, polymer brushes…” It is about structure, not shape.
5. The authors briefly reviewed the use of peptide for targeted drug delivery. However, several important peptide ligands, such as iRGD and CTX, are missed.
Author Response
To Reviewer #2:
Dear Reviewer,
First of all, we would like to express our deep gratitude for careful evaluation of our manuscript and your competent comments. We have thoroughly revised the manuscript according to your remarks. Please find below the detailed answers to the questions mentioned in your review (Reviewer’s remarks are presented in bold, our answers follow in plain text, quotations from the manuscript are shown in italic).
In this review manuscript by Kutova et al, the authors intended to provide an overview of the strategies that have been developed for targeted drug delivery to tumors. The authors reviewed the biology of tumors relevant to drug delivery, following with discussion of three different approaches. This is a potentially interesting review which may have a broad audience. The following critiques need to be addressed.
1. Emerging evidence shows that the EPR effect, which has been overemphasized based on findings in animal studies, has not been well validated in human patients. The authors need to review the relevant findings in human patients and discuss the potential reasons. This discussion may help distinct this review article from many others with a similar focus.
We add the discussion on the role of EPR-effect in human patients.
Though the multiple reports about successful EPR effect-based creative solutions in cancer treatment are expanding it should be noted, that data are accumulated on the absence of the expected effect when moving from animal models to human tumours [115,116]. The most evident example illustrating the gap between preclinical models and clinical practice is the clinical trials results for FDA-approved nano-formulations Doxil® (PEGilated liposomal doxorubicin) and Abraxane (nanoparticle albumin-bound (nab)-paclitaxel). The treatment efficiency of Doxil was comparable to free doxorubicin. Its clinical benefit resulted from the modified profile of side effects, primarily from reduced cardiotoxicity [117]. The second example, AbraxaneTM, in clinical trials was significantly more efficacious than conventional formulation Taxol. In this case, the emulsion of Cremophor® EL, a modified castor oil used for paclitaxel solubilization in Taxol, was eliminated, that resulted in significant increase of maximum tolerated dose [118]. In both cases, the benefits over conventional formulations can be credited to EPR-effect only to a small extent.
Currently, the inconsistency in drug distribution in animal tumor models and in tumours of human patients is considered to be a potential cause of such a discrepancy in the results of preclinical and clinical trials. The growth rate of a human tumour and its relative volume is much smaller than in case of an experimental model, at the same time an absolute volume is often much larger. Thus, the microenvironmental conditions in the human tumours have their peculiar characteristics: vascular network with fewer fenestrations, hypoxic and acidified sites formed due to the heterogeneity of the blood supply, heterogeneous basal membrane and reduced pericyte coverage, matrix heterogeneity and rigidity, elevated interstitial fluid pressure which promotes convective transport with subsequent nanomedicine clearance from the central zone of the tumour to its periphery [119,120]. The set of these parameters can vary considerably basing on individual characteristics of the patient, type of the tumour the patient suffers of and the stage of its development, hence the manifestation of EPR effect also differs [121].
2. In the Active targeting section, the authors reviewed “Cancer-specific molecular targets” in section 3.1. In addition to the molecular targets expressed on cell surface, molecular targets in tumor microenvironment are also important for drug delivery. The authors need to include a section to provide an overview of these targets.
By the term “cancer-specific molecular targets” we mean all the tumour targets, both stromal and expressed on the surface of tumour cells.
We extended the definition to avoid misunderstanding.
In order to precisely attack a tumour an appropriate target should be selected. A target molecule must be overexpressed on the surface of malignant cells compared to normal cells or, in ideal situation, be absent in normal tissues (Table 1). Also, the potential using of tumour stroma associated targets should be noted.
We also added the discussion on a diversity of the potential stromal targets. Actually, the majority of the stroma-targeted treatment approaches implies using agents that do not exhibit a specific tropism to tumour, but yet have a specific action towards them. The agents of this type as well as activatable and stimuli-responsive agents, which also demonstrate selective action at the tumour site, are beyond the scope of the Review (as is indicated in the Introduction section). We have, however, included the short discussion on the topic.
The above listed molecules are expressed on the surface of tumour cells. However, stromal components of a tumour, both nonmalignant cells and extracellular matrix (ECM), can also serve as a target for directional treatment. Tumour progression is strongly determined by the microenvironment of tumour cells and crosstalk between tumour cells, cancer associated fibroblasts, endothelial cells, pericytes and smooth muscle cells composing tumor vasculature, as well as infiltrating immune and inflammatory cells [213]. Targeting non-malignant cells of tumour stroma (including their signaling pathway participants) [214-216] and vasculature [217-219] seems to be the potent treatment approach with most remarkable progress in immune checkpoints inhibition [220].
To date, the existing treatment strategies are mostly aimed at improvement in blood perfusion, drug extravasation and tissue penetration by modulating of tumour stroma as well as at activation of the antitumour immune response [221,222]. However, the specificity of tumour microenvironment makes it potentially possible to develop the approaches for drug delivery based on targeting stromal compartment. For example, a number of antiangiogenic nanoparticle-based agents with receptor specific peptides have been reported [223,224].
3. Line 175, “of using synthetic particles is in the rational molecule…” The authors should be aware that liposomes are also synthetic particles.
We corrected the confusing sentence.
An advantage of using such type of particles is in the rational molecule design that allows high reproducible synthesis of the carriers with desired properties.
4. Line 180: “a shape diversity… organized as dendrimers, hyperbranched polymers, polymer brushes…” It is about structure, not shape.
We corrected the confusing sentence.
Besides the material-based diversity of the polymer nanoparticles, there is also a diversity in shape, size, surface charge and structural organization [88]. Thus polymer nanoparticles can be organized as dendrimers, hyperbranched polymers, polymer brushes, or form the micelles and micelle-like structures [89-93].
5. The authors briefly reviewed the use of peptide for targeted drug delivery. However, several important peptide ligands, such as iRGD and CTX, are missed.
We extended the discussion on the tumor targeting peptides and included description of both iRGD and CTX groups.
Internalizing RGD (iRGD) peptides with 9-amino acid cyclic motifs was shown to launch transcytosis of nano-carriers through the vascular endothelium. This allows delivering loaded nano-carriers to the tumours with non-leaky vasculature where the EPR based delivery strategy is not applicable. A significant improvement of drug delivery and hence antitumor efficacy has been shown for a wide variety of nano-carriers in multiple tumour models [260], both in case when the iRGD-peptide was chemically bound to the nano-carrier [261,262] or co-administered [263].
Among cell penetrating peptides chlorotoxins or CTX-like peptides (a group of peptides derived from the scorpion venom) is intensely studied as targeting agents for targeted drug delivery. CTX peptide was purified and characterized in early 1990s [264] and then was shown to specifically bind to glioma cells [265]. The further research in this area indicated that CTX-like peptides can be internalized both by the energy-dependent (endocytosis) and by the energy-independent way. It was also shown that CTX-like peptides have a tumour binding activity, but their molecular target is not yet specified; nevertheless chlorine channels, matrix metalloproteinase-2 and annexin A2 are considered as potential candidates. Currently, the use of CTX-like peptides is investigated for various types of tumors (mainly for brain tumors) for imaging, drug delivery and radiotherapy [266,267].
Round 2
Reviewer 2 Report
The authors well addressed most of my critiques.
Line 183: "Thus polymer nanoparticles can be organized as dendrimers, hyperbranched polymers, polymer brushes, or form the micelles and micelle-like structures". The expression is confusing. Hyperbranched polymers and polymer brushes cannot be classified as polymer nanoparticles. The authors may simply remove this sentence.
Author Response
Dear Reviewer,
First of all, we would like to express our deep gratitude for careful evaluation of our manuscript and your competent comments. We have thoroughly revised the manuscript according to your remarks. Please find below the detailed answers to the critique mentioned in your latest review (Reviewer’s remarks are presented in bold, our answers follow in plain text, quotations from the manuscript are shown in italic).
Line 183: "Thus polymer nanoparticles can be organized as dendrimers, hyperbranched polymers, polymer brushes, or form the micelles and micelle-like structures". The expression is confusing. Hyperbranched polymers and polymer brushes cannot be classified as polymer nanoparticles. The authors may simply remove this sentence.
We removed the confusing sentence.